# Assessment of Spatial Temporal Changes of Ecological Environment Quality: A Case Study in Huaibei City, China

**Ruihao Cui [1], Jiazheng Han [1] and Zhenqi Hu [1,2,*]**

1   School of Environment Science & Spatial Informatics, China University of Mining & Technology, Daxue Road 1#, Xuzhou 221116, China; ts21160111p31@cumt.edu.cn (R.C.); cumthjz@cumt.edu.cn (J.H.)
2   Institute of Land Reclamation & Ecological Restoration, China University of Mining & Technology (Beijing), Beijing 100083, China
*   Correspondence: huzq@cumtb.edu.cn

**Abstract:** Under the short-term economic development goal, the excessive exploitation of natural resources and the destruction of the ecological environment make the ecological environment of Huaibei cities increasingly fragile. This study constructed the Remote Sensing Ecological Index (RSEI) to evaluate the ecological environment change trend and its driving factors in Huaibei City from 2000 to 2020. The barycenter migration model was used to determine the RSEI spatial change trend, and the geographic detector was used to analyze the influencing factors of the RSEI value change. The results showed that: (1) the average RSEI value of Huaibei City generally fluctuates within the range of good and excellent grades. (2) The migration direction of the barycenter of RSEI is similar when the level of RSEI improves or decreases from 2000 to 2020, and the barycenter migration is most severe from 2005 to 2015. (3) The driving factors of RSEI change were population density (0.47) > land use (0.24) > slope (0.14) > precipitation (0.08) > temperature (0.04) > altitude (0.03). All the factors had interaction effects on the RSEI, mainly with nonlinear enhancement. (4) From 2000 to 2010, urban construction encroached on all kinds of land, which was the direct reason for the decline in ecological environment quality. From 2010 to 2020, the surge of water and meadow areas improved the ecological environment quality of Huaibei city. Therefore, reducing the expansion of artificial land, returning farmland to forests and meadows, wetland park construction, and other ecological protection measures are the keys to ensuring the sustainable development of regional social and economic development. This study can provide a reference and scientific basis for sustainable development strategy and ecological protection planning to improve the ecological environment quality of Huaibei City.

**Keywords:** Google Earth Engine; remote sensing ecological index (RSEI); barycenter migration analysis; landcover; GeoDetector

## 1. Introduction

Ecological environment quality [1,2] refers to an ecosystem's ecological and environmental factors suitable for human survival in a specific time or space. It is a comprehensive indicator of the social economy and natural environment combination. Increased urbanization has led to disasters, from soil erosion to biodiversity degradation, energy shortages, and air pollution. The key is scientifically judging the quality of the urban ecological environment and conducting visual analysis, the most important scientific and practical means of predicting impact factors and conducting the manual intervention.

In recent years, scholars have been exploring a comprehensive quantitative method for assessing the quality of the regional ecological environment, which is divided into three categories: (1) the Ecological Index (EI) of China's Eco-environmental Standard Status, (2) single-type indicators, and (3) the Remote Sensing Ecological Index (RSEI) proposed by Xu [3]. The Ecological Index has multiple indicators such as biological abundance,

vegetation coverage, water network density, land degradation, and environmental quality, but some of these indicators are difficult to obtain using visualization analysis. The Normalized Difference Vegetation Index (NDVI) is the most widely used single indicator; it has been adopted in various ecological studies. The Leaf Area Index (LAI) [4] is another commonly used vegetation index for environmental change monitoring and analysis. The Enhanced Vegetation Index (EVI) [5] is usually used to reflect changes in vegetation covers in ecological environments. Therefore, the RSEI model was widely [6–8] used as soon as it was proposed. The evaluation index can quickly, objectively, and efficiently obtain the change in ecological environment quality, and it has the advantages of data visualization, spatiotemporal analysis, modeling, and prediction.

In recent years, the RSEI has become more and more popular in particular urban areas such as Beijing-Tianjin-Hebei [9], Suzhou-Wuxi-Hangzhou [10], and Erhai Lake Basin [11], and it is often used in ecological environment quality assessments. Internationally, there are also studies on different cities in Europe [12] and Russia [13]. A significant amount of time and space data in these studies are often required. The huge amount of data and the complex data preprocessing and index calculation work are the essential factors restricting RSEI analysis. The advent of the Google Earth Engine (GEE) cloud platform has revolutionized this.

The GEE cloud platform is the most advanced comprehensive geographic information analysis and visualization platform [14]. It stores massive geographic databases and supports real-time results previews, solving local download, storage, and preprocessing inefficiencies. With this advantage, GEE has been widely used in mapping the range and changes of large-scale land cover [15] types, long-term dynamic monitoring [16], crop yield estimation [17], etc.

Over the past 60 years since its establishment, Huaibei City has supplied nearly one billion tons of raw coal to the whole country, providing energy for the industrial and urban development of East China and even China. However, coal mining has brought serious ecological and environmental problems [18,19], such as water pollution, biodiversity reduction, soil quality decline, etc., which seriously affect the sustainable development of the economy and society and restrict the high-quality development of cities. Therefore, it is necessary to evaluate and analyze the impact factors of ecological environment quality, providing a scientific decision-making basis for future sustainable development. Based on the above, the purpose of this study is to: (1) construct the RSEI efficiently by integrating multiple sensors such as Landsat TM/OLI based on the GEE platform; (2) monitor the temporal and spatial changes in ecological environment quality in Huaibei City from 2000 to 2020; (3) explore the spatial distribution characteristics of ecological environment quality changes in Huaibei City and analyze the influencing factors; (4) provide recommendations about applications and adaptations for planning, development, and policymaking towards a sustainable future.

## 2. Materials and Methods

### 2.1. Study Area

Huaibei City is located in the northern part of Anhui Province, between 116°23″~117°23″ east longitude and 33°16″~34°14″ north latitude (Figure 1). It is adjacent to Suzhou in the east and to Guoyang and Yongcheng County in Henan in the west.

In the construction and development of Huaibei City, to promote economic and social development more quickly, it continuously exploits and utilizes natural resources, destroys the ecological environment, and obtains short-term economic benefits. The increase in human disturbance activities has changed the landscape pattern, and the land use structure of the ecosystem in Huaibei City has triggered the crisis of water resources, mineral resources, and land resources and has seriously deteriorated the local ecological environment. China has started the "13th Five-Year" ecological, environmental protection planning policy, and Huaibei city also started the "14th Five-Year" ecological, environmental protection planning policy. It involves adhering to the requirements of ecological civilization construction and

sustainable development strategy, understanding the current situation of the regional ecological environment, and improving the present poor ecological environment by changing the development strategy. The core problem to be solved urgently is how to systematically and accurately understand the past ecological quality changes to guide future ecological restoration work.

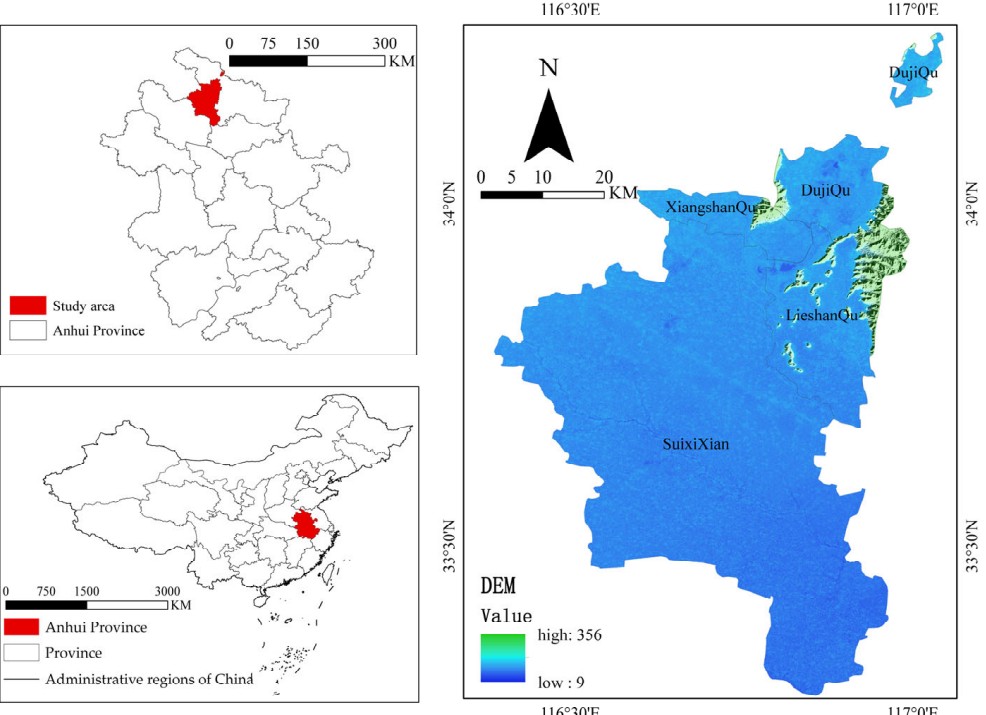

**Figure 1.** Location of the study area.

Using Huaibei City as an example, this paper uses the RSEI to represent eco-environmental quality. It reveals the laws and mechanisms of land use impact on eco-environmental quality by analyzing the effects of different driving factors on the RSEI, providing the theoretical basis for formulating sustainable development strategies for the ecosystem.

### 2.2. Data Acquisition and Preprocessing

The work of this study is represented by a flow chart (Figure 2). Firstly, the time-space analysis of the ecological environment quality of Huaibei City is carried out, and then the factors affecting the change in ecological environment quality are analyzed. (1) The Landsat images in 2000, 2005, 2010, 2015, and 2020 were obtained through the GEE platform, the four indices were calculated, and then the normalized principal component analysis was performed to obtain the RSEI grade map of Huaibei City. Superposition analysis was carried out on the RSEI grade maps to obtain the RSEI grade change maps from 2000 to 2005, 2005 to 2010, 2010 to 2015, and 2015 to 2020. The center of the barycenter shift of the RSEI grade in 20 years was plotted. (2) The three-phase land use data in 2000, 2010, and 2020 were obtained on the Globaland30 platform, and land use transfer analysis and land use change area analysis were performed on them. The land use change area map was one of the human factors in the geographic detector. Then, the influence factors were analyzed through the GeoDetector, and the heat map of the influence factors was drawn.

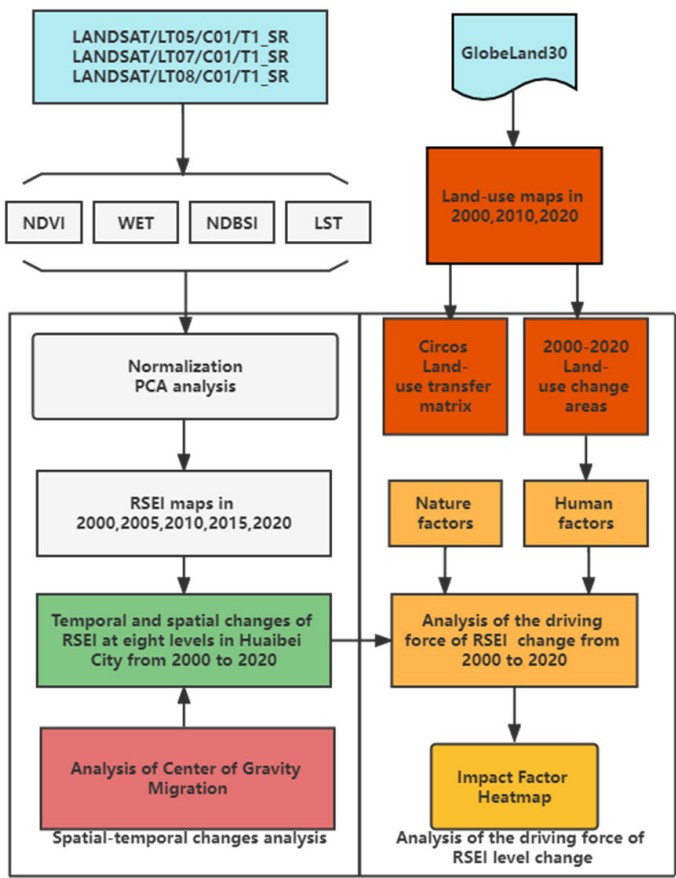

**Figure 2.** Workflow chart.

　　　The research data mainly include remote sensing, meteorology, land use, and population data (Table 1). The population data used in this paper come from the 2015 CnPop dataset [20], which is based on county-level census data and land use data at a scale of 1:100,000. Both the DEM data and Landsat data were downloaded using the GEE platform, with a spatial resolution of 30 m. The Landsat5, 7, and 8 surface reflectance datasets were accessed online through JavaScript APIs. Taking the Landsat5 surface reflectance dataset as an example, its images contain data after radiometric correction and atmospheric correction, and the pixel values of three visible bands (Band1(Blue), Band2(Green), Band3(Red)), the near-infrared band (Band4), and the short-wave infrared band (Band5) are corrected. In the thermal infrared band (Band6), the pixel value of surface reflectance (Band7) is the fixed temperature value at the sensor. Based on the GEE platform, we de-cloud and water mask the Landsat images; normalize the four ecological indicators of greenness, humidity, dryness, and heat; and use the principal component analysis method to construct the RSEI. The climate data come from WorldClim 2.1 [21], using the annual average rainfall and annual average temperature. The land use data come from the GlobeLand30 dataset [22] (spatial resolution is 30 m, overall accuracy is 85.72%, Kappa coefficient is 0.82), and three-year land use maps for 2000, 2010, and 2020 are drawn GlobeLand30 data includes 10 first-level types: cultivated land, forest land, grassland, shrub land, wetland, water body, tundra, artificial surface, bare land, glacier and permanent snow cover. Huaibei city only arable land, woodland, grassland, shrub land, water, artificial surface of the six types of land (Figure 3).

**Table 1.** Data Sources.

| Data Types | Year/Resolution | Data Source |
| --- | --- | --- |
| CnPop | 2020 (1000 m) | Data Sharing Platform of Earth System Science of National Science and Technology Infrastructure of China |
| DEM | 2015 (30 m) | https://earthexplorer.usgs.gov/, accessed on 26 March 2022. |
| Landsat5 SR | 2000–2013 (30 m) | https://developers.google.com/earth-engine/datasets/catalog/landsat-5?hl=en, accessed on 26 March 2022. |
| Landsat7 SR | 2013 (30 m) | https://developers.google.com/earth-engine/datasets/catalog/landsat-7?hl=en, accessed on 26 March 2022. |
| Landsat8 SR | 2013–2020 (30 m) | https://developers.google.com/earth-engine/datasets/catalog/landsat-8?hl=en, accessed on 26 March 2022. |
| GlobeLand30 | 2000–2020 (30 m) | www.globallandcover.com, accessed on 26 March 2022. |
| WorldClim 2.1 | 2020 (10 m) | http://worldclim.org, accessed on 26 March 2022. |

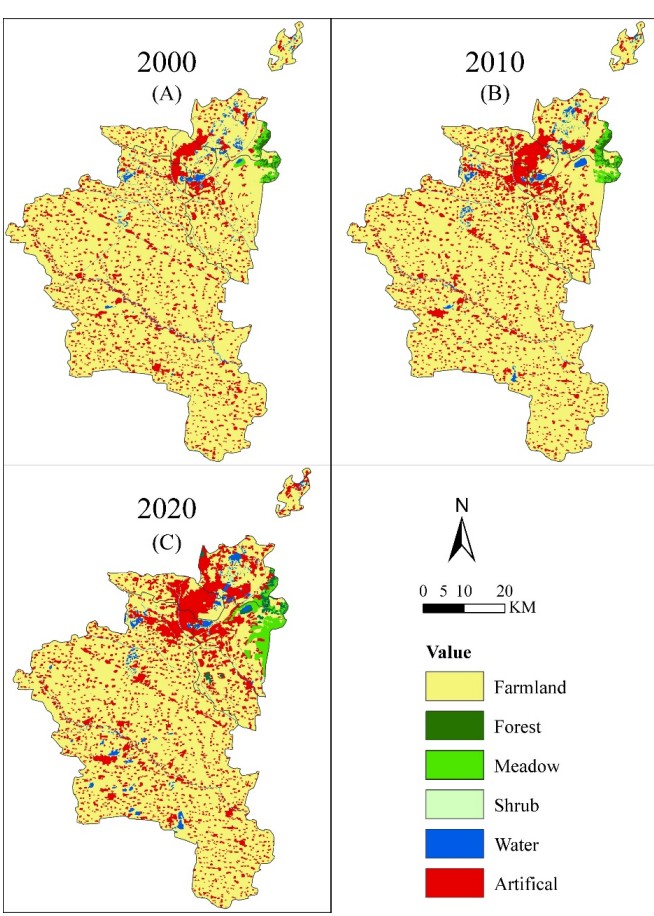

**Figure 3.** The land use map of Huaibei City from 2000 to 2020. The solid line in the figure is the administrative boundary line of Huaibei City.

*2.3. Method*

2.3.1. RSEI Calculation

The remote-sensing ecological index (RSEI) is a comprehensive index for rapidly detecting the eco-environmental conditions of a geographical region solely using remote-

sensing data. Four components (greenness, moisture, heat, and dryness) closely related to ecological environment quality were selected to construct the RSEI. The main information of the four indicators is mainly concentrated on the first principal component, enabling the RSEI to integrate the information of the four indicators.

(1) The humidity component in the tasseled [23] cap transformation represents humidity:

$$
\begin{aligned}
WET_{TM} &= 0.0315\rho_A + 0.2021\rho_G + 0.3102\rho_{NIR} - 0.6806\rho_{SWIR1} - 0.6109\rho_{SWIR2} \\
WET_{OLI} &= 0.1511\rho_B + 0.1972\rho_G + 0.3283\rho_{NIR} - 0.7117\rho_{SWIR1} - 0.4559\rho_{SWIR2}
\end{aligned}
\tag{1}
$$

where $\rho_B, \rho_G, \rho_R, \rho_{NIR}, \rho_{SWIR1}$, and $\rho_{SWIR2}$ are the reflectance data of the blue, green, red, near-infrared, short-wave infrared one, and short-wave infrared two bands of TM and OLI, respectively.

(2) The normalized vegetation index (NDVI) [24]: Vegetation is an essential factor reflecting the regional ecological quality. The greenness index adopts the normalized difference vegetation index (NDVI), representing the plant growth, vegetation density distribution, and vegetation coverage status.

$$
NDVI = (\rho_{NIR} - \rho_R) / (\rho_{NIR} + \rho_R)
\tag{2}
$$

(3) The normalized difference building-soil index (NDBSI) [25]: Soil drying caused by construction land and bare soil will seriously harm the region's ecological environment. Therefore, this paper uses two indexes—the index-based built-up (IBI) and the plain soil index (SI)—to calculate the NDBSI that represents the degree of soil drying:

$$
IBI = \frac{\{2\rho_{SWIR1}/(\rho_{SWIR1} + \rho_{SWIR2}) - [\rho_{NIR}/(\rho_{NIR} + \rho_R) + \rho_G/(\rho_G + \rho_{SWIR1})]\}}{\{2\rho_{SWIR1}/(\rho_{SWIR1} + \rho_{SWIR2}) + [\rho_{NIR}/(\rho_{NIR} + \rho_R) + \rho_G/(\rho_G + \rho_{SWIR1})]\}}
\tag{3}
$$

$$
SI = [(\rho_{SWIR1} + \rho_R) - (\rho_B + \rho_{NIR})] / [(\rho_{SWIR1} + \rho_R) + (\rho_B + \rho_{NIR})]
\tag{4}
$$

$$
NDBSI = (IBI + SI)/2
\tag{5}
$$

(4) The land surface temperature (LST) [26]: The heat index uses thermal infrared to represent the surface temperature. Whether on a global or regional scale, the thermal environment problem is a real problem that needs to be solved urgently. The surface temperature is calculated using the Landsat user manual model and revised parameters, and its expression is:

$$
LST = Tb/[1 + (\varepsilon\lambda Tb)/\rho] - 273.15
\tag{6}
$$

$$
Tb = K_2 / \ln(K_1/L_6 + 1)
\tag{7}
$$

$$
L_6 = \text{gain} \times \text{DN} + \text{bias}
\tag{8}
$$

$\lambda$ represents the wavelength of the Landsat thermal infrared band; $\rho = 1.438 \times 10^{-2}$ m; $\varepsilon$ is the emissivity obtained according to the NDVI threshold [27] processing proposed; $K_1$ and $K_2$ are the parameters obtained from the image source data; Tb is the brightness formula; DN is the gray value of the data pixel; gain and bias are the band gain value and offset value, respectively; and $L_6$ represents the radiation value of the Landsat thermal infrared band.

(5) RSEI calculation. Xu [28] used principal component transformation to construct a comprehensive remote sensing ecological index. The leading information of the four indicators is mainly concentrated on the first central component (PC1), which enables the RSEI to integrate the knowledge of the four indicators. Each band has different units and value ranges, so the four bands must be normalized separately. The formula is:

$$
Nli = (li - l_{\min}) / (l_{\max} - l_{\min})
\tag{9}
$$

*Nli* is the index normalization processing result; *li* is the *i*th pixel value; $l_{min}$ is the minimum value; and $l_{max}$ is the maximum value. The final formula is $RSEI_0$ = 1-PC1 [f (NDVI, WET, NDBSI, LST)]. When PC1 has a low value in areas with good ecological quality and a high value in areas with poor environmental quality, PC1 needs to be subtracted from one so that the high value of $RSEI_0$ represents the good ecological condition.

### 2.3.2. Analysis of the Center of Barycenter Migration

The center of the barycenter is a physical concept representing the location of the physical force balance point. This concept [29] can also be introduced in geography to describe the spatial "bias" of environmental factors or human influences [30]. The calculation formula is as follows:

$$
\begin{aligned}
\overline{X} &= \sum_{i=1}^{n} X_i Z_i / \sum_{i=1}^{n} Z_i \\
\overline{Y} &= \sum_{i=1}^{n} Y_i Z_i / \sum_{i=1}^{n} Z_i
\end{aligned}
\tag{10}
$$

The value of a certain point in space represents the geographic abscissa and ordinate at the point. Then, the finally obtained barycentric coordinates migrate to their relative positions in the study area. In this paper, the improvement and reduction of RSEI levels in different time periods are selected to analyze the temporal and spatial trend of RSEI.

### 2.3.3. Geographical Probe

Using Geodetector, the driving factors of ecological environment quality changes in Huaibei City from 2000 to 2020 were identified and analyzed. Proposed by Wang et al. [31], Geodetector is a new statistical method based on the spatial changes of geographic units. Geodetector was initially proposed to detect the risk of neural tube defects. In recent years, Geodetector has been used to detect geographic factors, land use changes [32], and others. An analysis via Geodetector can detect associations between dependent variables and their influencing factors and find dominant factors to quantify the interaction between two variables. The model contains four formulations: the factor detector, risk detector, interaction detector, and ecological detector. Not only can a GeoDetector model handle categorical and numerical dependent variables and find dominant factors, but the model can also quantify the interaction effect between two variables. If an independent variable *X* is associated with a dependent or response variable *Y*, *Y* will vary in space and time and present a similar distribution to *X*. The more substantial the similarity between *X* and *Y*, the more sensitive *X* is to *Y*. Therefore, the q-value is used to measure the similarity between *X* and *Y*. The following equation represents the q statistic:

$$
q = 1 - \left( \sum_{h=1}^{L} N_h \sigma_h^2 \right) / N\sigma^2
\tag{11}
$$

$$
\sigma^2 = \sum_{h=1}^{N_h} \left( R_i - \overline{R} \right)^2 / N
\tag{12}
$$

$$
\sigma_h^2 = \sum_{h=1}^{L} \sum_{j=1}^{N_h} \left( R_{h,j} - \overline{R_h} \right)^2 / N
\tag{13}
$$

where *N* refers to the total number of sample units in the entire study area, and $\sigma^2$ represents the global variance in *Y* in the entire study area. In the GeoDetector factors model, the study area is stratified into *L* zones ($h$ = 1, 2, 3, ... , *L*), and the stratification depends on the characteristics of the explanator or determinant factors (*X*). $N_h$ and $\sigma_h^2$ represent the number of sample units and the variance in *Y* within zone *h* considering fact *X*. $R_i$ is the value of the *i*th sample unit from the entire study area, and $\overline{R}$ represents the global mean

value of $X$. $R_{h,j}$ is the value of the $i$th sample unit of $X$ in zone $h$, and $\overline{R_h}$ is the mean value of $X$ in zone $h$. Thus, $\sum\limits_{h=1}^{L} N_h \sigma_h^2$ is the sum of the variance within the zone.

The interaction detector has a solid ability to detect spatial heterogeneity and can detect and calculate the interaction between two other factors. In addition, $q(X_i \cap X_j)$ is the decisive force for exchanging the two factors, which can reveal whether, together, the two factors $X_i$ and $X_j$ enhance or weaken the explanation of $Y$ relative to their independent effects. The interaction types are shown in Table 2.

**Table 2.** Definition of the interaction types in the GeoDetector model.

| Interaction Relationship | Interaction Types | Description |
|---|---|---|
| $q(X_i \cap X_j) < \text{Min}(q(X_i), q(X_j))$ | Nonlinear weaken | The interaction of two variables nonlinearly weakens the impacts of single variables. |
| $\text{Min}(q(X_i), q(X_j)) < q(X_i \cap X_j) < \text{Max}(q(X_i), q(X_j))$ | Univariable weaken | The impacts of single variables are univariably weakened by the interaction of two variables. |
| $q(X_i \cap X_j) = q(X_i) + q(X_j)$ | Independent | The impacts of single variables are independent. |
| $\text{Max}(q(X_i), q(X_j)) < q(X_i \cap X_j) < q(X_i) + q(X_j)$ | Bivariable enhanced | The impacts of single variables are bivariably enhanced by the interaction of two variables. |
| $q(X_i \cap X_j) > q(X_i) + q(X_j)$ | Nonlinear enhanced | The interaction of two variables nonlinearly enhances the impacts of single variables. |

This study uses geographic detectors to identify and analyze the driving factors of the ecological environment index in Huaibei City from 2000 to 2020. Both natural and human factors may affect the quality of the ecological environment, and various factors may interact. Therefore, this study uses the factor detector to study the factors that affect the chances of the ecological environment index and uses the interaction detector to determine the strength of the interaction between the two elements.

In this study, two factors affecting human activities (land use ($X_1$) and population density ($X_2$)), two meteorological factors (temperature ($X_3$) and precipitation ($X_4$)), and two topographic factors (slope ($X_5$) and altitude ($X_6$)), were selected as the impact factors put into the geographic detector to explore the main influencing factors. Since Geodetector's requirements for independent variables are categorical variables, independent variables need to be discretized and classified. This study uses the natural breakpoint method to classify population density, temperature, precipitation, slope, and altitude into eight categories. Then, it uses the 2000–2020 land use map (Figure 3) to draw the land use transition map and classify it into 24 categories according to the changes in the types of features as the initial data of the Geodetector. This study divided Huaibei City into a regular grid of 100 m × 100 m. Then, the data between the independent and dependent variables can be connected by sampling in each grid.

## 3. Results

### 3.1. Ecological Grade Analysis

The following can be seen in Figure 4: (1) NDVI representing greenness and WET representing humidity are positive values, while LST representing heat and NDBSI representing dryness are negative values, consistent with the actual situation. (2) Different land types of regional ecological environment quality are different. For example, the ecological environment quality in desert areas is 0.2 [33], it is 0.5–0.7 in urban [34,35] agglomerations, it is 0.5 in the Loess Plateau [36,37], it is 0.2 [38] in wetlands, and it is as high as 0.83 in farmland [39] areas. The 20-year average RSEI of Huaibei City is 0.76, which is closest to the regional RSEI value represented by farmland. The RSEI value in 2000–2020 was in the range of 0.6–0.9, but there was a great fluctuation in 2007–2008; that is, the RSEI value decreased from 0.91 to 0.59, which may be related to human factors such as land use change, climate change, rainfall impact, and interference from natural factors.

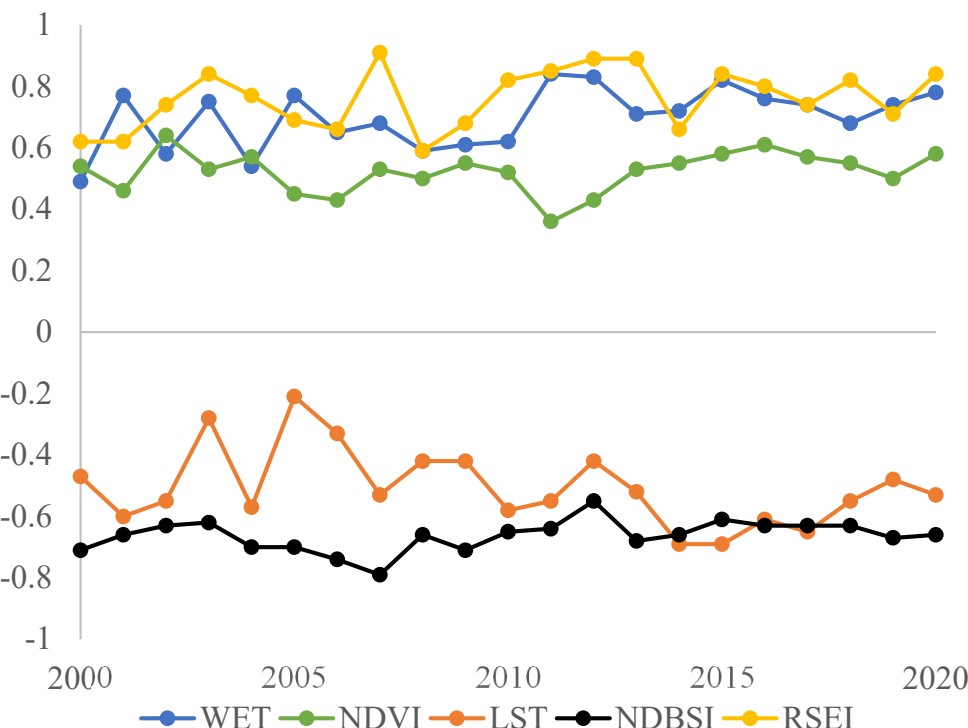

**Figure 4.** 2000−2020 RSEI and four component size scatter plots.

To further quantify and visualize the RSEI, the calculation results are divided into five grades with an interval of 0.2: excellent (0.8~1), good (0.6~0.8), moderate (0.4~0.6), fair (0.2~0.4), and poor (0~0.2), as shown in Figure 5. Each grade is combined with the Technical Specifications for Evaluation of Ecological Environment Conditions (HJ192~2015) [40]. Poor means that the ecological environment conditions are relatively poor and that human life is restricted; fair standards mean that the vegetation coverage is poor, and obvious factors are slowing human life; moderate means that the vegetation coverage is moderate, which is more suitable for human habitation, but there are some constraints on human life, indicating a non-ideal living environment; good means that the vegetation coverage is high and fit for human life; excellent means that the vegetation coverage is high and the ecosystem is stable. From Figure 5, it can be seen that the ecological environment quality of Huaibei City from 2000 to 2020 was at a good grade. Still, the ecological environment quality at the junction of Xiangshan District, Lieshan District, and Suixi County was at a poor or fair grade for a long time. The statistics on the area and the proportion of each grade in 2000, 2005, 2010, 2015, and 2020 are shown in Table 3.

We calculated the sum of the proportions of poor, fair, and moderate grades (PFM%) and the sum of good and excellent grades (GE%) to judge the overall ecological environment quality of Huaibei City. In 2000, 2005, 2010, 2015, and 2020, the PFM% and GE% were 6.61%/93.39%, 30.45%/69.55%, 4.92%/95.08%, and 2.18%/97.82%, respectively, indicating that it increased and then decreased, while the GE% decreased first and then increased. During the whole research period, the area of Huaibei City that was in the good grade decreased yearly. The most significant reduction in the area was 605.9 square kilometers between 2000 and 2005 (Table 4); the area in the excellent grade first remained unchanged and then increased yearly. The most significant area increase was 998.46 square kilometers; the proportion of the area in the moderate grade increased from 5.83% in 2000 to 29.04% in 2005 and then returned to its original appearance in the following years.

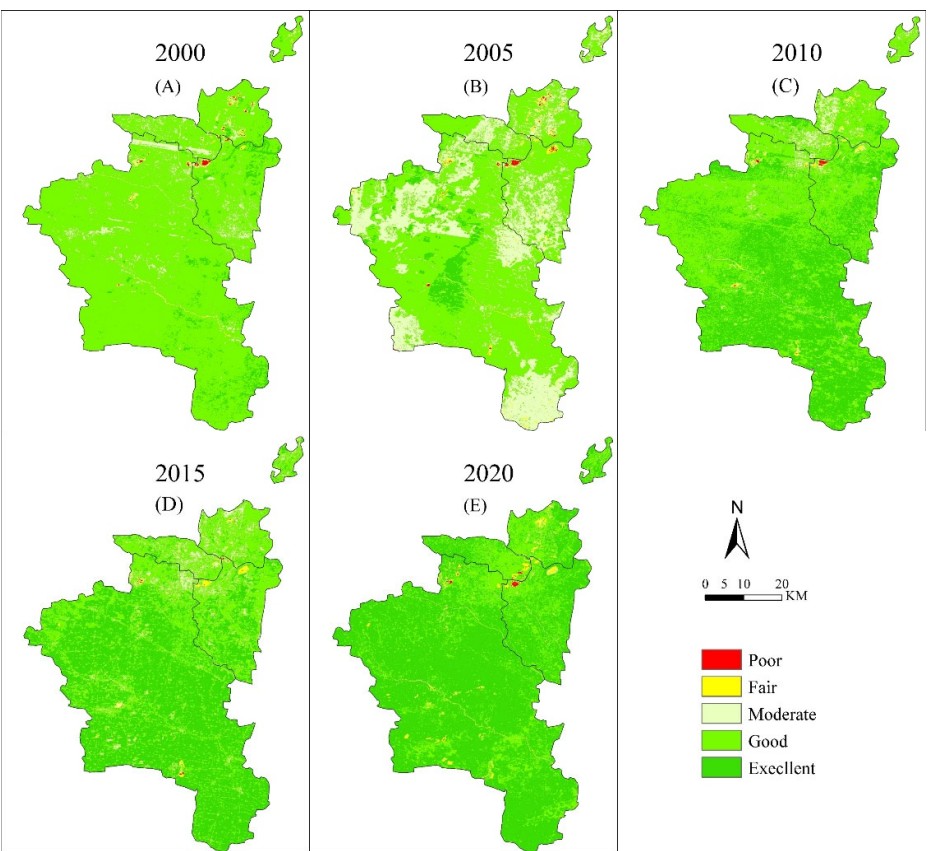

**Figure 5.** Areas of different RSEI quality grades in Huaibei City during 2000–2020. The solid line in the figure is the administrative boundary line of Huaibei City.

**Table 3.** Area statistics of changes in the RSEI grades in Huaibei City during 2000–2020.

| RSEI Grade | 2000 | | 2005 | | 2010 | | 2015 | | 2020 | |
|---|---|---|---|---|---|---|---|---|---|---|
| | Area (km²) | Pct (%) | Area (km²) | Pct (%) | Area (km²) | Pct (%) | Area (km²) | Pct (%) | Area (km²) | Pct (%) |
| Poor/(0–0.2) | 5.07 | 0.19 | 6.20 | 0.23 | 2.30 | 0.09 | 0.04 | 0.00 | 2.85 | 0.11 |
| Fair/(0.2–0.4) | 14.70 | 0.55 | 31.50 | 1.18 | 13.98 | 0.52 | 5.49 | 0.21 | 24.65 | 0.92 |
| Moderate/(0.4–0.6) | 155.00 | 5.83 | 776.44 | 29.04 | 115.25 | 4.31 | 28.73 | 1.07 | 30.83 | 1.15 |
| Good/(0.6–0.8) | 2380.00 | 89.00 | 1744.1 | 65.24 | 1428.30 | 53.43 | 1022.80 | 38.26 | 668.62 | 25.01 |
| Excellent/(0.8–1) | 117.3 | 4.39 | 115.14 | 4.31 | 1113.60 | 41.65 | 1616.30 | 60.46 | 1946.40 | 72.81 |

Based on the division of the above five grades, the difference change detection of the RSEI in each year in Huaibei City was carried out. Green represents areas where the quality of the ecological environment has improved, and yellow represents areas where the ecological environment has deteriorated. It can be seen from Figure 6 that the ecological grade transition between the southeastern and northern regions of Huaibei City is more obvious. The southwest area of Huaibei City had a high degree of change in the ecological grade between 2000–2005 and 2005–2010 (Figure 6A,B), and there was a significant change in 2010–2015 and 2015–2020 (Figure 6C,D). The northern part of Huaibei City, especially the Duji District, has a large area of ecological grade transfer, and the overall ecological quality first decreased and then improved (greener). However, at the intersection of Duji District, Lieshan District, and Xiangshan District, it has been in a state of ecological deterioration (only from 2005 to 2010 did the ecological grade rise) for a long time. To better understand the changes in ecological grades, this study used ArcGIS software to analyze the area transfer of ecological grades in Huaibei City from 2000 to 2020 (Table 4).

**Table 4.** RSEI transfer matrix in Huaibei City during 2000–2020. (unit: km$^2$).

| 2000 | 2010 | | | | | |
|---|---|---|---|---|---|---|
| | **Poor** | **Fair** | **Moderate** | **Good** | **Excellent** | **Total** |
| Poor | 161 | 34 | 20 | 21 | 0 | 236 |
| Fair | 235 | 410 | 237 | 540 | 11 | 1433 |
| Moderate | 89 | 754 | 1924 | 8759 | 287 | 11,813 |
| Good | 32 | 290 | 9974 | 131,332 | 4651 | 146,279 |
| Excellent | 2 | 20 | 3430 | 103,551 | 7112 | 114,115 |
| Total | 519 | 1508 | 15,585 | 244,203 | 12,061 | 273,876 |

| 2010 | 2020 | | | | | |
|---|---|---|---|---|---|---|
| | **Poor** | **Fair** | **Moderate** | **Good** | **Excellent** | **Total** |
| Poor | 139 | 54 | 13 | 73 | 12 | 291 |
| Fair | 67 | 619 | 640 | 854 | 347 | 2527 |
| Moderate | 9 | 267 | 1346 | 1251 | 290 | 3163 |
| Good | 18 | 426 | 7451 | 42,100 | 18,223 | 68,218 |
| Excellent | 3 | 67 | 2363 | 102,001 | 95,243 | 199,677 |
| Total | 236 | 1433 | 11,813 | 146,279 | 114,115 | 273,876 |

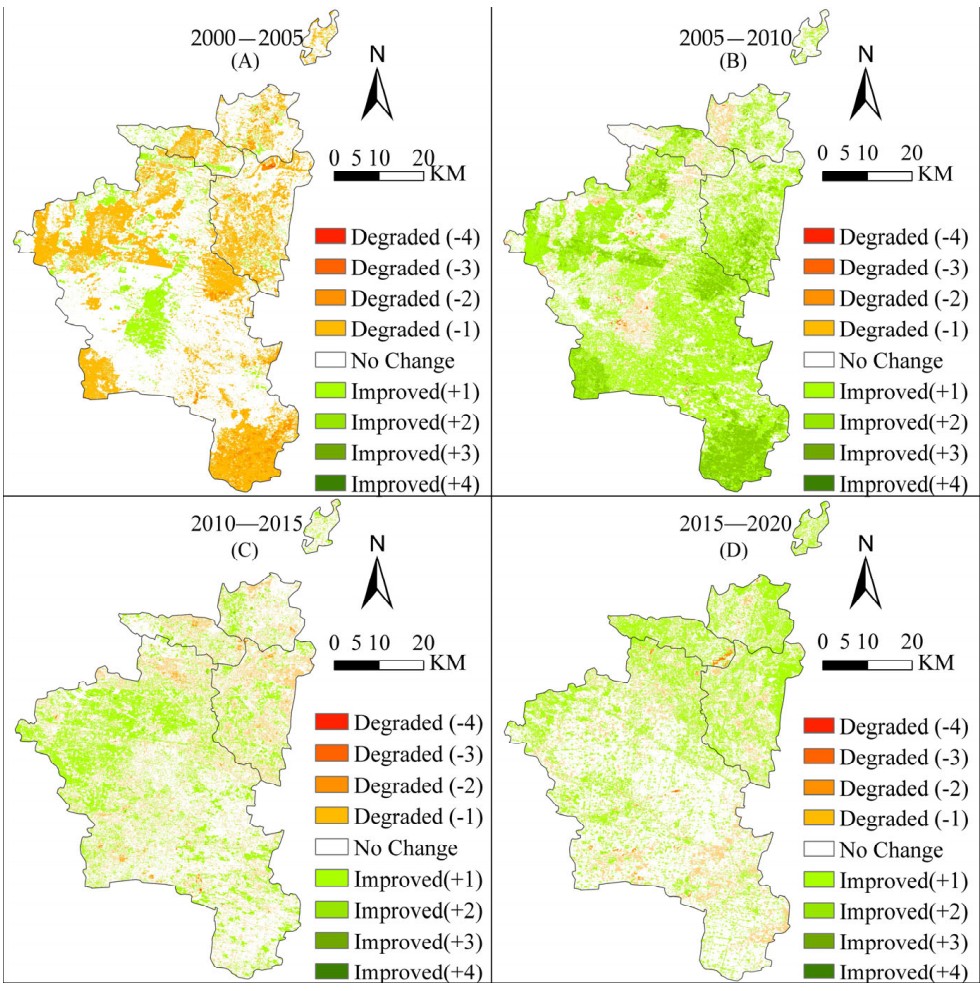

**Figure 6.** Spatiotemporal changes in the RSEI in Huaibei City during 2000–2020. The change level is +4, +3, +2, +1, −1, −2, −3, and −4. "+" represents the improvement of the ecological level, and "−" represents the deterioration of the ecological grade. "+4" means that the RSEI grade of a given area is represented by a pixel increased by four grades, e.g., from Poor to Excellent.The solid line in the figure is the administrative boundary line of Huaibei City.

From 2000 to 2010, 43.26% of the land ecological quality grades declined, and only 5.28% of the land ecological quality grades improved. The area converted from GE% to PFM% is 14,148 km, accounting for 5.15%, and the area converted from PFM% to GE% is 9618 km, accounting for 3.5%. From 2010 to 2020, the ecological quality grade of 112,999 km of land increased by 41.15%, and the ecological quality grade of 8.07% of the land decreased. Among them, the area converted from GE% to PFM% was 10,326 km, accounting for 3.76%, and the area converted from PFM% to GE% was 2827 km, accounting for 1.02%. Therefore, it can be concluded that the overall ecological environment quality of Huaibei City tends to be stable.

### 3.2. Analysis of Barycenter Migration

To further understand the spatial change trend of ecological environment quality in Huaibei City, the barycenter migration model was used to calculate the barycenter coordinates of different RSEI trends, and the barycenter migration diagram of the RSEI grade changes from 2000 to 2020 was drawn based on each barycenter position (Figure 7). From the perspective of the migration trajectory of RSEI grade changes from 2000 to 2020, the trajectory of the barycenter shows a similar pattern: the center of gravity first moves to the south and then moves to the northeast from 2005 to 2020. From 2000 to 2010, the barycenter of the RSEI grade increased to the southeast, while the barycenter of the RSEI grade decreased to the opposite direction.

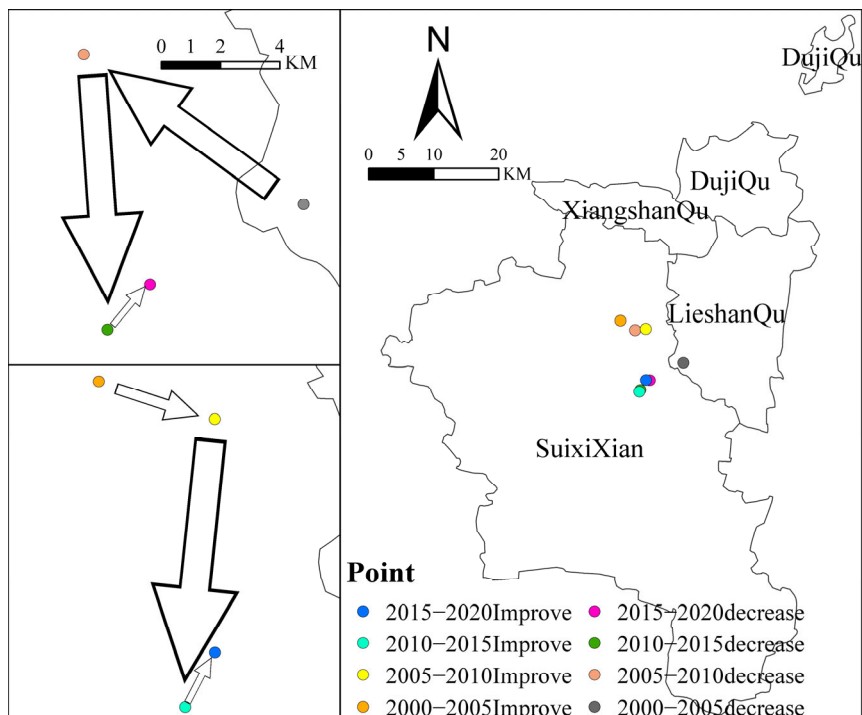

**Figure 7.** The barycenter migration map of the RSEI change curve in Huaibei City.

From the perspective of distance, it can be seen that the spatial position change of the RSEI grade decrease is relatively drastic, with a total movement of 19.27 km in 20 years, and the maximum migration distance between two points is 9.25 km. The spatial position change of the RSEI was slightly gentle, with a total migration of 15.32 km in 20 years and a maximum migration distance of 9.83 km. As seen from Figure 7, the spatial distribution of the center of gravity points is relatively discrete, indicating that there is no large-scale change in ecological environment quality, and the migration of the ecological environment level in Huaibei City shows a relatively stable trend.

### 3.3. Land Use

Table 5 shows the change trend and the area of land use in Huaibei City from 2000 to 2020. During the whole research process, the area of farmland and shrub forests showed an almost linear downward trend, and the speed gradually increased. The area of farmland and shrubland showed an almost linear downward trend, and the decline rate gradually accelerated. During 2000–2010 and 2010–2020, the area of farmland decreased by 2.17% and 6.81%, and the area of shrubs decreased by 0.10% and 0.28%. From 2000 to 2010 and from 2010 to 2020, the meadow area increased by 0.28% and 1.65% respectively. The artificial surface increased by 2.12% and 4.82%. In the past 20 years, the urbanization of Huaibei City has developed rapidly, which inevitably leads to the occupation of farmland and forests, which also makes the area of meadows expand several times. The area of the water showed a trend of first decreasing and then increasing, with an overall increase of 0.24%. The forest area increased first and then decreased, with an overall increase of 0.16%. In the past 20 years, the main change in Huaibei City has been the mutual conversion of farmland and other land use types. The area transferred from farmland to other types was 39,164.21 km$^2$, of which 76.34% was converted to artificial land and 9.13% was converted to meadows.

**Table 5.** Land use type area transfer matrix (unit: km$^2$).

| 2000 | 2010 | | | | | | |
| --- | --- | --- | --- | --- | --- | --- | --- |
| | **Meadows** | **Shrubs** | **Farmland** | **Artificial** | **Forests** | **Water** | **Total** |
| Meadows | 1101.4 | 0 | 23.29 | 0.59 | 49.3 | 0 | 1174.58 |
| Shrubs | 0 | 0 | 112 | 1.4 | 0 | 98.32 | 211.72 |
| Farmland | 96.07 | 156.55 | 212,514.2 | 13,082.34 | 975.73 | 1575.28 | 228,400.17 |
| Artificial | 3.03 | 16.88 | 7714.79 | 28,603.73 | 13.78 | 119.8 | 36,472.01 |
| Forests | 48.37 | 0 | 94.38 | 13.91 | 1660.95 | 269.38 | 2087.35 |
| Water | 1.52 | 10.53 | 1991.02 | 577.27 | 68.49 | 2881.55 | 5530.38 |
| Total | 1250.39 | 183.96 | 222,449.68 | 42,279.24 | 2768.61 | 4944.33 | 273,876.21 |
| 2010 | 2020 | | | | | | |
| | **Meadows** | **Shrubs** | **Farmland** | **Artificial** | **Forests** | **Water** | **Total** |
| Meadows | 997.81 | 2.15 | 29.34 | 27.11 | 193.98 | 0 | 1250.39 |
| Shrubs | 0 | 20.33 | 2.97 | 0.5 | 0 | 160.16 | 183.96 |
| Farmland | 3477.78 | 22.71 | 199,171.44 | 16,816.34 | 747.67 | 2213.74 | 222,449.68 |
| Artificial | 207.54 | 2.88 | 3604.82 | 38,207.69 | 59.45 | 196.86 | 42,279.24 |
| Forests | 1082.23 | 6.91 | 130.4 | 150.4 | 1249.23 | 149.44 | 2768.61 |
| Water | 9.85 | 53.56 | 860.69 | 283.52 | 270.06 | 3466.65 | 4944.33 |
| Total | 5775.21 | 108.54 | 203,799.66 | 55,485.56 | 2520.39 | 61,868.5 | 273,876.21 |

The land use change from farmland to other types was more pronounced than the land use change from other types to farmland. Figure 8 shows the expansion and occupation of a large area of urban construction land and the water accumulation of small cultivated land in Huaibei city from 2000 to 2010. It can be seen that the area of cultivated land occupied by construction land decreased, and the large area of vegetation increased. The areas where the farmland is artificial are all located in the northern region. The area of farmland converted to an artificial surface in the north is much larger than that in the south, which may be related to the impact of urbanization in the north. The changes in shrubs and meadow are mainly concentrated in the eastern ridge area of Lieshan District, which is related to the Regulations on Returning Farmland to Forests issued by the Huaibei Municipal Government in 2008. The changes in the water area are concentrated in Duji District and the southwest of Suixi County. According to the Huaibei City Wetland Protection and Development Plan, the original mining subsidence land was transformed, and the wetland was formed.

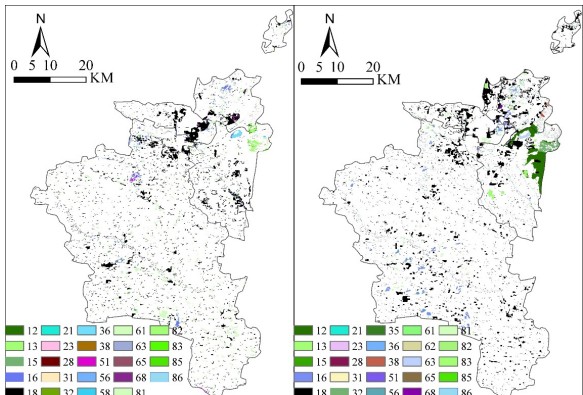

**Figure 8.** 2000–2020 land use change areas (number "1" represents meadows, number "2" represents shrubs, number "3" represents farmland, number "4" represents artificial land, number "5" represents forests, number "6" represents water, number "13" represents the conversion of shrubs to farmland, and so on).

The change in land use will have a particular impact on the region's ecological environment, and the evolution of the environment will directly affect the production and life of human beings. Changes in land use will have two different impacts on the ecological environment: one is a positive impact, that is, the quality of the domain is improved, such as returning farmland to forests, land reclamation, and the construction of artificial wetlands; the other is a negative impact, that is, the deterioration of the original ecological environment, such as reclaiming the land around lakes, disorderly occupation, development of the original cultivated land for production and construction, etc., which will lead to the reduction of vegetation in the region and the loss of soil and water, which will lead to the deterioration of the ecological environment.

*3.4. Analysis Results of Geographic Detector Driving Factors*

The ecological environment quality problem is caused by human activities on the natural environment intervention. In order to find a way to improve the quality of the ecological environment, this paper selects 6 factors such as population, land use, precipitation, slope and temperature to calculate the relative impact (q statistic); the calculation results are shown in Table 6. The larger the q statistic, the stronger the influence the factor has on the change in ecological environment quality. The population density influences the RSEI $(q(X_2) = 0.47)$, which is the dominant factor. The influences of land use and slope, as the main factors, were both higher than 0.1 $(q(X_1) = 0.24, q(X_5) = 0.14)$. Temperature, precipitation, and altitude are secondary factors. Ecological environment quality is a systematic problem in which all factors are inseparable, so it is necessary to analyze the interaction between the influencing factors. As can be seen from Figure 9, the pairwise factor interaction q statistic is larger than the corresponding single factor interaction q statistic. A total of 20 pairs of bifactors exhibited nonlinear enhancement, and pairs of factors exhibited bivariate enhancement $(X_2 \cap X_1)$. The highest q statistic of interaction between paired factors is that of population density and slope, which is 0.88, indicating that topographic factors and population density are highly correlated with ecological environment quality. The worst explanatory power is the q value of the temperature and altitude, which is 0.17. Although the effects of the slope, temperature, precipitation, and altitude on the ecological environment quality are relatively weak, their explanatory power is significantly enhanced when combined with population slope and land use. For example, this trend can be seen in the interaction of temperature and slope $(q(X_3 \cap X_5) = 0.35, q(X_3) = 0.04, q(X_5) = 0.14)$, precipitation and population density $(q(X_4 \cap X_2) = 0.84, q(X_4) = 0.08, q(X_2) = 0.47)$, and so on.

**Table 6.** Factor detection results.

| | Land-Use Change ($X_1$) | Population Density ($X_2$) | Temperature ($X_3$) | Precipitation ($X_4$) | Slope ($X_5$) | Altitude ($X_6$) |
|---|---|---|---|---|---|---|
| q-statistic | 0.24 | 0.47 | 0.04 | 0.08 | 0.14 | 0.03 |

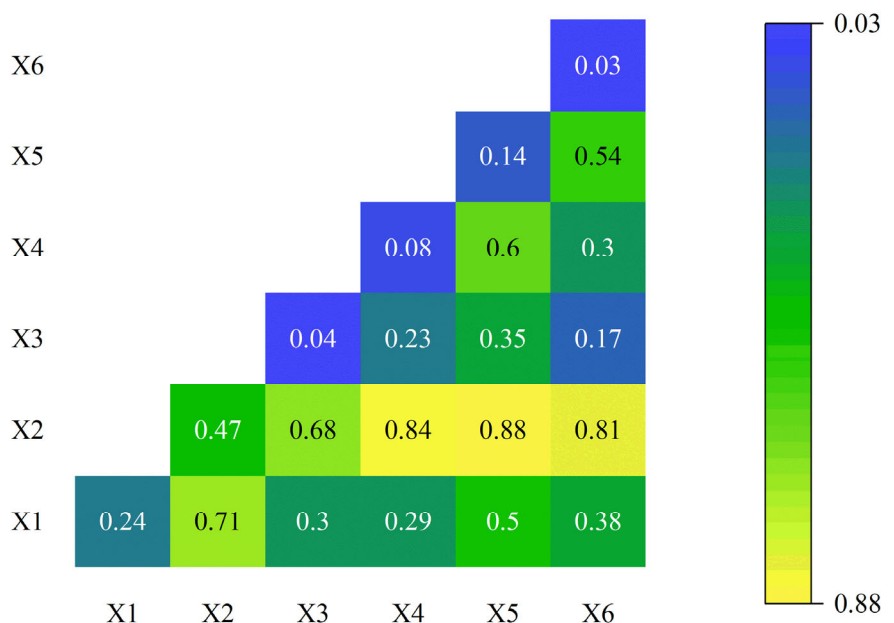

**Figure 9.** Interaction detection results.

## 4. Discussion

### 4.1. Applicability of Evaluation Indicators and Sustainable Development Goals

Ecological sensitivity analysis can also be used to evaluate ecological environment quality. Yang et al. [41] used the ecological sensitivity index method based on the pressure sensitivities and elasticity (PSR) model to analyze the ecological environment quality in Huaibei City. The conclusion shows that most of the area (70.51%) of Huaibei City is in a slightly fragile state; that is, the ecological environment is in a state of good quality, and the ecologically extremely fragile land is mainly distributed in the northern urban center and the southwest of Huaibei City. This conclusion is consistent with that shown in Figures 5 and 6. This proves the credibility of the RSEI in evaluating ecological environment quality. The application of the RSEI to the SDG 15 may consist in two aspects. (1) The RSEI contains indicators for the construction [42] industry, which can be roughly applied to the impact of the sustainable built environment on biodiversity conservation. (2) The RSEI perfectly covers indicators of terrestrial [43,44] ecosystems and can be used to replace the enhanced vegetation index and water network density in the biodiversity indicator system in the sustainable development strategy.

### 4.2. Suggestions for the Future Ecological Environment Improvement of Huaibei City

Based on the results of the ecological grade analysis and driving factor analysis, some suggestions for the future ecological environment protection in Huaibei City were put forward:

(1) The change in land use type is the main factor of the RSEI grade change, so the government should pay more attention to the impact of land use type change on farmlands and artificial lands and conduct good land use planning when carrying out ecological environment protection work. (2) The central area of the northern part of Huaibei City is planned as a key ecological protection area for key restoration. (3) The Huaibei Municipal

Government can use the RSEI to conduct a simple assessment of sustainable development goals to accelerate the transformation and development of resource-depleted cities.

### 4.3. High Groundwater Level Coal Mining in Huaibei City

Since the 1950s, the coal mining subsidence area has been produced and has begun to take shape, but the coal mining volume was small, and the subsidence area grew slowly, with an annual increment of 1–2 km$^2$. According to statistics [45,46], the subsidence land area of every 10,000 t of coal mined in the Huaibei Coalfield is about 3200 m$^2$. Since the 1980s, the amount of coal mining has gradually increased, and the growth rate of the subsidence areas has also accelerated. The total surface subsidence area of the Huaibei Coalfield has reached 171 km. Part of the subsidence area has accumulated water, formed lakes, and connected into pieces, with an area of about 51 km. The deepest subsidence center is 22m. These had seriously harmed the mining area's living environment and affected the ecological environment index. Huaibei City is where plain cultivated land and coal resources overlap, and coal mining will inevitably destroy cultivated land. In addition to the shallow groundwater location [47,48], the surface subsidence field caused by coal mining has seasonal water accumulation throughout the year, which affects the physical, chemical, and biological properties of the soil; affects the growth of crops [49]; and causes varying degrees of damage to cultivated land. The groundwater level [47,50,51] after mining and subsidence easily rises, causing a large amount of high-yield and high-quality cultivated land with perennial or seasonal water accumulation to reduce or stop production, which seriously affects the production and life of farmers. Coal and farmland environments play crucial roles in energy security and the ecological environment, respectively. However, coal mining has destroyed farmland, deteriorated the ecological environment, and created a conflict between humans and land, which hinders the sustainable development of the ecological environment.

### 4.4. Advantages and Limitations

The novelty of this study is that, based on the GEE cloud platform, we analyze the spatial changes of the ecological environment quality in Huaibei City and analyze the possible changes in the barycenter; use GeoDetectors to determine the driving factors that affect the ecological environment; and make a reasonable assessment of the future ecological environment changes in Huaibei City. Although this paper has determined some of the driving factors that affect the quality of the ecological environment, it does not consider the impact of the high-water-level coal mining data and mining area distribution data on the ecological environment in Huaibei City and lacks relevant analysis.

## 5. Conclusions

In the context of global efforts to achieve the SDGs, this study can provide an assessment tool for the transformation and sustainable development of resource-exhausted cities. In this paper, the GEE platform was used to extract the remote sensing ecological environment index, and the ecological environment quality of Huaibei City from 2000 to 2020 was monitored. On this basis, the changing trend of the RSEI grade was examined by the barycenter migration model. The factors affecting the change in ecological environment quality were explored through the geodetector, the impact of land use on RSEI change was mainly analyzed, and the following conclusions were drawn:

(1) The overall ecological environment quality of Huaibei City showed a zigzag fluctuation trend, and the fluctuation range slowed down. Only at the intersection of Duji District, Lieshan District, and Xiangshan District was the ecological environment quality grade low for a long time.

(2) Land use and slope are the main factors, while temperature, precipitation, and altitude are secondary. The interaction between various factors can enhance the influence of the RSEI, and the interaction between population and topography is the most significant.

(3) Land use has a relatively high impact on ecological environment quality. The change of farmlands and artificial lands from other land types directly leads to the difference in ecological environment quality. From 2000 to 2010, urbanization intensified the occupation of artificial meadows and different land types, which was the main reason for the decline in ecological environment quality. From 2010 to 2020, the area of meadows and water increased dramatically, and the urbanization rate slowed down, improving ecological environment quality.

Future research should quantitatively analyze the contribution rate of different factors to the RSEI change in the study area, carry out a prospective scenario prediction study of the RSEI in Huaibei City based on various population migration backgrounds, objectively reveal its change law, better serve regional ecological planning, and promote the harmonious development of man and nature.

**Author Contributions:** Conceptualization, R.C.; methodology, R.C.; software, R.C.; validation R.C.; formal analysis, R.C.; investigation, R.C.; resources, R.C.; data curation, R.C.; writing—original draft preparation, R.C.; writing—review and editing, J.H. and R.C.; visualization, R.C.; supervision, Z.H.; project administration, Z.H.; funding acquisition, Z.H. All authors have read and agreed to the published version of the manuscript.

**Funding:** This study is funded by the Jiangsu Province University Innovation Team Project (Grant number 20192036) and the Jiangsu Province University Innovation Talent Project (Grant number 20191468).

**Institutional Review Board Statement:** Not applicable.

**Informed Consent Statement:** Not applicable.

**Data Availability Statement:** The data are available from the corresponding author upon reasonable request.

**Acknowledgments:** We thank the anonymous reviewers for their constructive comments on the earlier version of the manuscript. All individuals agree to confirm.

**Conflicts of Interest:** The authors declare no conflict of interest.

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
