# Peer review of "Assessment of Spatial Temporal Changes of Ecological Environment Quality: A Case Study in Huaibei City, China"

_land, doi:10.3390/land11060944_

Round 1

Reviewer 1 Report

In their manuscript submitted to the mdpi/Land Special Issue "The Eco-Environmental Effects of Urban Land Use" for consideration for publication the authors perform an environmental analysis for Huaibei City, China using a range of indices, most notably the Remote Sensing Environment Index (RSEI), and compare results over a 20-year time period to describe dynamics as well as the impact of selected parameters on the overall index. Based on their work, they achieve to document the development and the improvement of the Huaibei city area, and they also achieve some separation and quantification of various controlling reasons that they briefly discuss. 

The work presented here is a straightforward technical workflow investigating and analysing change dynamics as well as a range of triggers and effects on the environmental quality using remotely sensed data from which indices are built. While some indices are simple and quite common, the authors include approaches that have found some attractiveness in a national context over the years, but are less commonly used in international context. This, however, is feasible and valid, if some background is provided and if the results are discussed against other approaches. 

Due to the lack of that, and due the lack of a richer discussion and in particular conclusion which presents some more international aspects, I feel that this manuscript has only regional relevance and it does not even try to disseminate and attract a wider readership beyond national boundaries. This is a major shortcoming for a manuscript submitted to an international journal, and it becomes quite clear when reading about the study area which somewhat expects deeper insights from the readership into the local settings. 

The authors also fail to communicate implications and avoid providing recommendations in how far this investigation can be transported to other scenarios and in how far this type of investigation could potentially help to improve policy development. 

Given these issues, along with a number of additional clarifications that would be needed and which are listed below, I would recommend major revision. The presentation quality, however, is absolutely sub-standard and although it is not justified to reject a manuscript based on that, it adds to quite a negative impression

Specific Comments:

Title: Your assessment is at no time based on GEE. GEE is a platform you are using. Your assessment is based on remote sensing indices and a national model that you incorporate. Please re-discuss the title internally.

(1) Please send your manuscript to a language editor/proof-reading service (or change it if you already have). There a plenty of issues, including grammar, wording, and incomplete sentences. It seems to have been translated from original Chinese when I look at some structural issues, but I am sure you can judge that better.

------------

(2) The introduction reads in principle well, but it sets a stage without providing much context. 

L30. You write as if everyone should know "Ecological Environment Quality" as a fixed term. Not everyone does, so please introduce the concept and the respective published papers before you do.

L54: "widely used" is not supported by references. Provide some, not only national ones.

L60: provide supporting references.

L70: check reference

L85: and perhaps you can add another purpose as outlined above: "(4) Provide recommendations about applications and adaptations for planning and development as well as policy-making towards a sustainable future".

------------

(3) The section on Material and Methods lacks a few things to make that presentation complete. 

(3.1) While initial methods and materials are introduced, this section does not communicate how the analysis is performed. The reader learns about that while reading the results. I would suggest to make the analysis approach clear in the methodology, including your statistical methods. 

(3.2) This also includes how you assess the validity of your results, as you have not mentioned that at all. Any accuracy assessment? I believe that this is quite important but unfortunately missing.

(3.3) Figure representation lacks some quality control, I have to say. First of all, be aware that not everyone knows where Chinese provinces are. Start your communication on a higher level, please. The map is in terrible shape. Colours, alignments, font sizes, scale bar and north arrow sizes, inset figures... everything yells "we do not care".  I am sorry to be so direct, but with all the tools that we have -- and you are using a professional GIS here -- there is no excuse in producing something that is in such a terrible shape. Apart from the presentation, why should anyone be interested in a two-colour terrain model representation? Create a hillshade to show the regional relief and superimpose colours to show gradients. Please look at some examples and improve this display. 

L90: An introduction of a study area is not about showing its location. It is about discussing its development in the context of this investigation. In particular when it comes to industrial changes that are responsible for the change of you ecological framework. There might also have been a local policy that introduced new control and guidelines, so the reader would like to read about it. Otherwise this whole investigation is just a bunch of some analyses without context. And that does not merit publication.

L97:  The workflow does not communicate the analysis part. I would suggest to add that.

L102: "It has better mapping performance and spatial heterogeneity details" -- better than what? What does that mean?

L105: Communicate which bands you want to work with and which are available, and foremost, tell the reader why.

L110: Sentence does not make sense, incomplete.

L116: Please do some formatting here, this is not acceptable. Spaces before opening parentheses, spaces before units, spaces between "Landsat" and "8" and so on. Please be more careful how you present material.

L123: Likely wrong reference here? Please check.

L129: The whole part on formulas yells carelessness and lacks any quality. Make sure you are using consistent indices and symbols ACROSS your formulas. When you use indices, make sure they are indeed indices, and not on the same line (e.g., formula 9). 

L152: Provide proper sources for your formulas, also when you cite the Landsat Handbook.

L161: Xu (YEAR)

L167: Unclear what the authors want to say here.

L171: The purpose of this method has not been communicated. Why is this method chosen? What is the aim? It feels ambiguous and arbitrary as it all reads like a list of methods without providing underlying purpose for the selection.

L184: Completely unclear. Before diving into the theory, write about purpose and aims. This whole section up to 221 needs to be rewritten accordingly.

------------

(4) Results are to large extent ok. However, they depend on what was introduced in the methodology and here not everything is clear. So, I am cautious saying that it is ok.
Please improve the methods and then I will be happy to re-read the results.

L238: I doubt that three subsections are needed in you document. It does not help to make things clearer. Please consider cleaning things up a bit.

L240: Wouldn't you believe that a scatter plot would be helpful to better see how values change over the different domains? Also, why you think that the mean of means is needed in the last line? Given the heterogeneous and unbalanced distribution of land use in the city, this value has no purpose, has it?

L256: Check reference

L298: (also the other figure, L371): introduce the diagram visualization and discuss it appropriately. Also, "Sankey" is the linear version. For circular diagram it is called "Circos", I believe. But I am sure you will double check.

L301: You use "ha" in the Sankeys and "sqkm" in tables. While not higher maths, it is still inconsistent if you change units.

L312: Figure 6 does not show years in the figure, only in the caption. I suggest to make it consistent and present years in the figure as well.

L338: The purpose of the CoG "analysis" remains a mystery to me after all. What is the purpose? What does it help to visualize? I am not saying you need to remove it, but I would like to read a clear methodological description along with a purpose, and a good discussion.

L365: I would suggest to alter the image and add relevant labels into the legend. Better even to remove all unused land classes. This presentation does not help.

L391: Diagram and underlying statistics has never been introduced or explained. Please correct hat and discuss results accordingly. Otherwise it is just another arbitrary figure.

------------

(5) The Discussion and Conclusions are quite incomplete, I feel. 

(5.1) The critical assessment of result, also in the context of other research in similar fields would be helpful. What about accuracy, for example? What about feasibility of such studies (effort, resources)... 

(5.2) It would also be good to communicate some judgment about the effort and suitability of such an approach as demonstrated -- in particular when it comes to developing local/regional policies. How could local authorities benefit? What are your recommendations? "We recommend effective
management in RSEI change areas and key land-use types, emphasizing anthropogenic disturbances due to population numbers." is not really helpful.

(5.3) I am missing the link to the Sustainable Development Goals and their metrics. This would be not only an optional, but a highly needed addition. This would also establish a link to a more international platform.

Author Response

Thank you for your research, we have done the revision

Author Response

thank

Reviewer 3 Report

Dear Authors, the paper related to spatio-temporal changes in ecological environment quality in a selected area of China may serve as an example of an index-based evaluation approach with the main target to monitor environmental and ecological sustainability and management directions affected by natural and anthropogenic factors. Minor changes in the manuscript are needed, for example, abbreviations should not be used in the title of the paper. The text in several places should be rewritten as it is hard to read due to a large number of used abbreviations. For better clarity, it is suggested to create a table of all abbreviations used and their decipherment at the beginning of the article. Furthermore, the description of formulas should be reviewed, as some of them have variables not described clearly. Line 162 needs an indication of reference. The overall structure of the paper can be also improved. The references to all tables and figures have to be reviewed, as it was noticed that, e.g., Figures 8 and 9 are not referred to in the text. The list of references needs appropriate style revision.

Author Response

Thank you for the opportunity, I have done the revisions, especially the abbreviations

Round 2

Reviewer 1 Report

Dear authors,

thank you for addressing my comments in detail and for introducing a number of changes that made -- I believe -- your presentation much clearer and which also contributed to provide more needed context information, in particular with respect to the local development plan.

Your current presentation is more coherent and changes regarding supporting figures helped to improve understanding. The methodology is presented more consistently and the new workflow also helped understanding what you did.

There are still some odd wordings/structures and ambiguous meanings (e.g., "The overall ecological environment of Huaibei city showed a trend of zigzag fluctuation and general rise." -- what does rise mean here?), so I would suggest to make second round of improvements here to make your statements clearer.

The link to SDG is still quite weak, but at least it was briefly discussed. I would suggest to add a conclusion, i.e., a link to SDG by formulating a recommendation regarding SDG. Give it a try, I am sure you will find a link to embed your research into a larger context.

Kind regards.

Author Response

感谢您对论文的评论

Reviewer 2 Report

- reference error in line 69
-Please check "visualisationtion word in line 48
- Empty character in line 15
-I would change Table 1 column title “Data Accuracy” as “Year/Resolution”

-line 136 “lansat 5” should be “landsat5” i think
-line 321 – figure 7A, 7B should be Figure 6A or 6B I think
-line322 – Fig. 7C-7D should be Fig.6C-6D I think
-I can’t read Table 4. It is not in English– line331

-Figure 8 shows some land-use changes from class 8 to class 1 but in the legend, there is no class 8. It is complicated to understand Figure 8 I think. The authors may show important changes.

Author Response

Thank you for your comments on the paper
